# Implementation of clinically relevant and robust fMRI-based language lateralization: Choosing the laterality index calculation method

**Irène Brumer**[1,2], **Enrico De Vita**[1]*, **Jonathan Ashmore**[2,3], **Jozef Jarosz**[2], **Marco Borri**[2]

**1** Department of Biomedical Engineering, School of Biomedical Engineering and Imaging Sciences, King's College London, London, United Kingdom, **2** Department of Neuroradiology, King's College Hospital, London, United Kingdom, **3** Department of Medical Physics and Bioengineering, NHS Highland, Inverness, United Kingdom

* enrico.devita@kcl.ac.uk

**Data Availability Statement:** The reason why the underlying Data cannot be shared publicly is explained below. Subjects were recruited, and informed written consent was obtained, with

## Abstract

The assessment of language lateralization has become widely used when planning neuro-surgery close to language areas, due to individual specificities and potential influence of brain pathology. Functional magnetic resonance imaging (fMRI) allows non-invasive and quantitative assessment of language lateralization for presurgical planning using a laterality index (LI). However, the conventional method is limited by the dependence of the LI on the chosen activation threshold. To overcome this limitation, different threshold-independent LI calculations have been reported. The purpose of this study was to propose a simplified approach to threshold-independent LI calculation and compare it with three previously reported methods on the same cohort of subjects. Fifteen healthy subjects, who performed picture naming, verb generation, and word fluency tasks, were scanned. LI values were calculated for all subjects using four methods, and considering either the whole hemisphere or an atlas-defined language area. For each method, the subjects were ranked according to the calculated LI values, and the obtained rankings were compared. All LI calculation methods agreed in differentiating strong from weak lateralization on both hemispheric and regional scales (Spearman's correlation coefficients 0.59–1.00). In general, a more lateralized activation was found in the language area than in the whole hemisphere. The new method is well suited for application in the clinical practice as it is simple to implement, fast, and robust. The good agreement between LI calculation methods suggests that the choice of method is not key. Nevertheless, it should be consistent to allow a relative comparison of language lateralization between subjects.

## Introduction

In the majority of individuals, language functions are predominantly located in the left brain hemisphere [1]. Nevertheless, language centers are usually spread over both hemispheres and

ethical approval from the UK National Research Ethics Service (REC 04/Q0706/72). Under the UK Health Research Authority (HRA) system the Research Ethics Committee do not directly regulate access to data, and should not be approached about such issues. Data access lies under the overall governance of the sponsoring organisation - and ultimately depends on what the subjects have consented to. The information sheet given to the healthy volunteers participating to this study states 'We may share the images we collect with others involved in the research (including researchers at other hospitals or universities, or working for the funding bodies)'. Although we may be able to ask for this in future studies, for the data reported here we do not have the consent from the participants to a) upload the images to a repository b) make the images publicly available c) share the images with researches not involved in this research (and we do not have permission to recontact the participants about this). Therefore, unfortunately, the fMRI raw images and the calculated images (statistical parametric maps) cannot be provided on this occasion. Nonetheless, we do completely appreciate the importance of making all the raw numerical data used in the paper available. We have therefore provided all LI values calculated for each subject using the four presented methods, and also the resulting subject rankings, as supplementary data (S1 Table). We have also provided the curveLI 'mean' and '95% confidence interval' curves for all tasks and ROIs shown in Fig 4 as data points (supplementary data S2 Table). These define and summarize our reference cohort, and can be used for future single subject or cohort comparisons. All these raw data constitute the results from this research.

**Funding:** This work was carried out at, and supported by, the Department of Neuroradiology at King's College Hospital NHS Foundation Trust. EDV is supported by the Wellcome/EPSRC Centre for Medical Engineering [WT 203148/Z/16/Z]. The views expressed are those of the authors and not necessarily those of the NHS, the NIHR or the Department of Health. The funders had no role in study design, data collection and analysis, decision to publish, or preparation of the manuscript.

**Competing interests:** The authors have declared that no competing interests exist.

vary from individual to individual in both their location and extent [2]. In addition, hemispheric or regional dominance in language functions (language lateralization) has been shown to depend on several factors such as age, handedness [3,4] and varies when evaluated in different brain regions [5,6]. More importantly, brain pathology can alter intra- and interhemispheric brain functionality [7], potentially causing a reorganization of language centers. Such a reorganization has been shown to lead to a change of language laterality from left to right hemisphere in left temporal lobe epilepsy [8–11], explaining the higher occurrence of atypical language lateralization (i.e. not left dominant) in epilepsy patients [12]. A similar change of language laterality has also been demonstrated in patients recovering from a stroke [13], and in brain tumor patients [4]. For these reasons, the assessment of language lateralization has become widely used when planning neurosurgery close to language areas. The information obtained from such assessment is useful for estimating the risk of post-surgical deficits [14], selecting an appropriate surgical approach [15,16], and deciding on the resection extent [16,17,18]. Direct electrical stimulation and the intracarotid (sodium) amobarbital test or Wada test [19] have often been used to assess language lateralization. However, these methods are invasive and can lead to complications and irreversible disabilities [20,21].

Functional magnetic resonance imaging (fMRI) can also be used for the assessment of language functions and has the advantage of being non-invasive, quantitative and suitable for pre-surgical planning. The assessment of language lateralization using fMRI involves the comparison of signals obtained during rest and activation phases of a task performed by the subject during the scan. This signal comparison is performed at the voxel level using a statistical test and produces activation maps, known as statistical parametric maps (SPMs). Hemispheric or regional dominance in language functions can then be quantified employing the laterality index (LI), which indicates the prevalence of activation in one side of the brain over the other [17,22]. The LI is conventionally calculated using the following formula:

$$LI = (NL - NR)/(NL + NR) \tag{1}$$

where *NL* and *NR* are the number of voxels with value above a specific activation threshold in left and right region of interest (ROI), respectively. LI values thus range from +1 (left dominant) to -1 (right dominant). However, a major limitation of this approach is the strong dependence of the LI on the arbitrarily chosen activation threshold value [23–25].

To overcome this limitation, various threshold-independent laterality index calculation methods have been reported in the literature [25]. For example, Knecht and colleagues define the activation threshold by a fixed total number of voxels and from this calculate a single LI value [26]. Abbott and colleagues expanded this idea and proposed to calculate the LI as a function of the total number of activated voxels and produce a LI curve, which can be used to visually evaluate a patient's lateralization compared to a healthy control group [24]. Matsuo and colleagues proposed to calculate a global LI by averaging conventional LI values evaluated over a range of thresholds [27]. Branco, Suarez and colleagues proposed to integrate the weighted histogram of voxel counts against threshold in order to calculate a global LI [23,28]. The review by Bradshaw and colleagues [25] highlights the lack of standardization in assessment of language lateralization using fMRI. The multitude of LI calculation methods, tasks and ROIs used in previously published reports makes comparison of results between studies difficult. Furthermore, the implementation of these methods is not always straightforward, which limits the feasibility of adoption in the clinical routine. When wishing to implement the assessment of language lateralization using fMRI in the clinical routine, a choice has to be made regarding the method. A direct comparison between threshold-independent LI

calculation methods using the same subjects, tasks, and ROIs could provide useful insights to aid this choice.

In this work we 1) implement and compare three previously reported threshold-independent methods applied to the same set of fMRI datasets from healthy volunteers, and 2) propose a simplified approach to threshold-independent LI calculation (*AUCLI*), which we compare to the other methods. Lateralization is evaluated on both a hemispheric and regional scale using three different language tasks.

## Methods

### MRI sequence protocol

Fifteen healthy right-handed volunteers (age range 21–45, mean ± standard deviation = 29 ± 7, 12 female) were scanned at 1.5 T (Magnetom Aera, Siemens AG, Erlangen, Germany) using the standard 20-channel head-only receive coil. Informed written consent was obtained with ethical approval from the UK National Research Ethics Service (REC 04/Q0706/72). The MRI sequence protocol consisted of a 3D T1-weighted MPRAGE anatomical sequence (TE/TR = 3.02/2200 ms, voxel = $(1 \text{ mm})^3$, FA = 8˚, parallel imaging acceleration of 2) and three BOLD contrast fMRI gradient echo EPI sequences (TE/TR = 40/3000 ms, voxel = 2.5x2.5x3 $\text{mm}^3$). For a given language paradigm, the fMRI protocol consisted of 6 cycles of alternating rest and activation periods of 30 seconds, resulting in a total scan time of 6 minutes (120 measurements).

### fMRI paradigms

The stimuli consisted of black letters or drawings on a white background and were presented visually to the subject using a screen at the end of the scanner bed, visible via a set of mirrors positioned on the head coil. Each volunteer performed three different language tasks: verb generation, picture naming, and word fluency. For verb generation, nouns appeared on the screen (15 per cycle) and the subject had to silently generate verbs associated with the noun. For the picture naming task, line drawings appeared on the screen (10 per cycle) and the subject had to silently name the depicted object. For the word fluency task, letters appeared on the screen (7 per cycle) and the subject had to generate words starting with the presented letter. For each task, the stimuli were randomly chosen from a pool of nouns, pictures, or letters. During the resting periods, a black cross-hair on a white background was projected in the center of the screen. The language tasks were set up using SuperLab 4.0 (Cedrus, San Pedro, California, USA).

### Image processing

The fMRI data was processed with the software package SPM12 (Wellcome Trust Centre for Neuroimaging, University College London, UK) using an in-house developed batch processing pipeline, which included following steps: 1) Images within each fMRI dataset were realigned to the first image (rigid body spatial transformation and least square algorithm, SPM12) to compensate for small degrees of motion. 2) The fMRI data was then co-registered to the anatomical data (non-linear mutual information registration algorithm, SPM12) and 3) smoothed using an isotropic Gaussian kernel with 8 mm full width at half maximum. SPMs were then calculated within the general linear model framework using a Student's t-test with a family-wise error rate significance level set at 0.05. In the following, the voxel values of the calculated SPM are referred to as threshold values or t-values. For all LI calculations, only voxels

with positive t-values are considered as these indicate activation correlating with the performed task.

## Regions of interest

In this work, the LI was calculated using both hemispheric and regional ROIs. For the definition of the ROIs, different brain atlases available in FSL (Wellcome Centre for Integrative Neuroimaging, FMRIB Analysis Group, University of Oxford, UK) were employed. The high resolution T1-weighted Montreal Neurological Institute (MNI) standard brain was used as reference and was mapped to the T1-weighted MPRAGE acquisition using both affine and non-linear registrations (FSL FLIRT [29–31] and FNIRT [32,33], respectively), thus allowing the direct transformation of atlas structures from the standard space to the acquisition space. Left and right hemisphere ROIs encompassing the entire cortical hemispheres (excluding the cerebellum) were created from the anatomical structures defined in the Harvard-Oxford Subcortical Structural Atlas [34–37]. The language ROIs encompassed Broca's area (Brodmann's areas 44 and 45) and Wernicke's area (posterior division of the superior temporal gyrus) as defined by the Jülich Histological Atlas [38–41] and Harvard-Oxford Cortical Structural Atlas [34–37]. The ROIs defined in the acquisition space for one of the subjects can be seen in Fig 1.

## Threshold-independent laterality index calculation methods

All LI calculations were performed using in-house software developed in MATLAB (Version 2016a, The MathWorks, Inc., Natick, Massachusetts, USA).

**1. Fixed total number of activated voxels (curveLI).** The method reported by Abbott and colleagues [23], labeled as curveLI, relies on the fact that each threshold corresponds to a total number of activated voxels. The curveLI method plots the LI values calculated for each threshold using Eq (1) against the total number of activated voxels. The total number of activated voxels considered for the calculations ranged from zero (no voxel within the ROI) to the maximum (all voxels with positive value within the ROI). In order to plot the curves obtained for different subjects in a single graph, the total number of activated voxels was normalized to the maximum for each subject, bringing the x axis in the range 0 to 1. This compensated for differences in ROI size from subject to subject due to individual brain anatomies. The distributions from all subjects were then used to calculate a mean LI and the 95% confidence interval (CI) of this mean:

$$curveLI^{mean} = \left(\sum curveLI^{individual}\right)/n \tag{2}$$

$$95\% \ CI = \frac{1.96}{15} * \sqrt{\frac{\sum \left(curveLI^{individual} - curveLI^{mean}\right)^2}{n-1}} \tag{3}$$

Where curveLI^individual denotes the LI calculated for a single individual, and n denotes the total number of subjects (n = 15 in this study). A single LI value was extracted at a threshold value corresponding to half the voxels being active (mid curve) for comparison with the other LI indices.

**2. Average (AveLI).** Following the method reported by Matsuo and colleagues [26], LI values were calculated for each t-value existing within the ROI considered, using Eq (1). These individual LI values were then averaged to form a global index:

$$AveLI = \sum_{t_{min}}^{t_{max}} LI(t\_value)/N \tag{4}$$

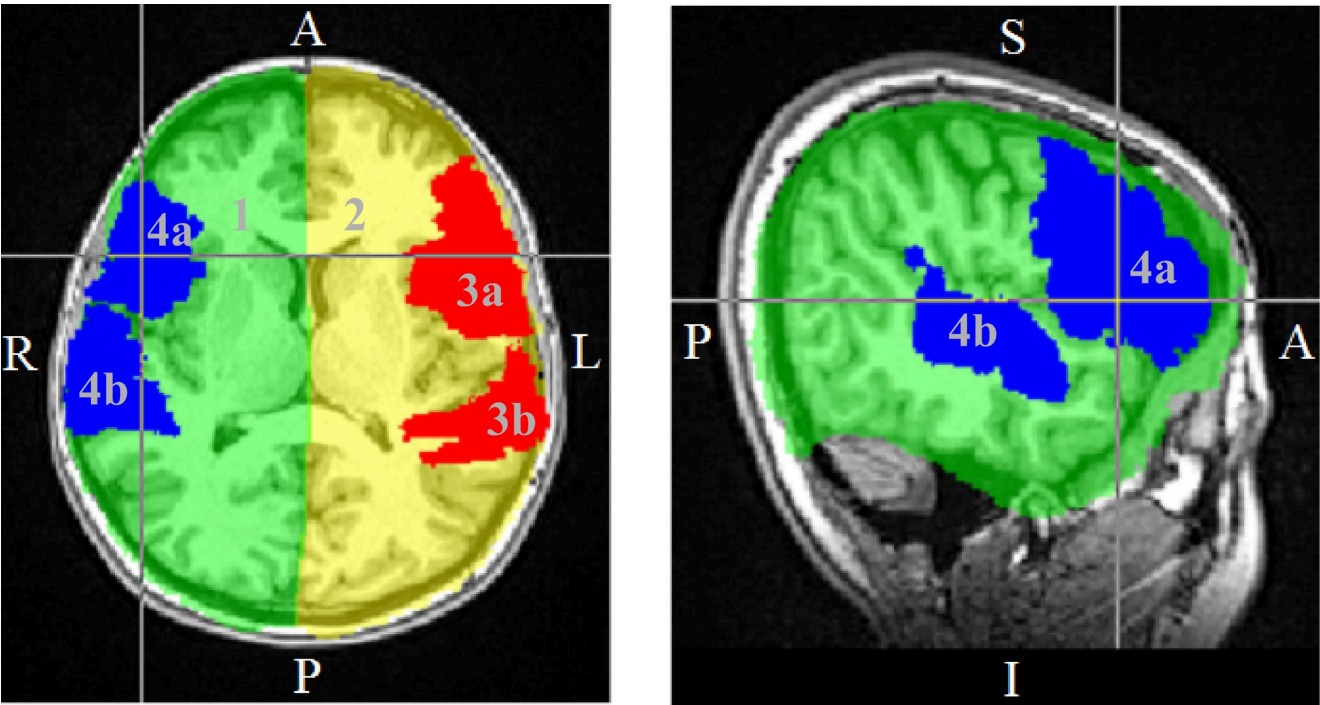

**Fig 1. Regions of interest defined using standard brain atlases.** 1 and 2 are the right and left *hemisphere ROIs*, 3 and 4 are the left and right *language ROIs* with 'a' designating Broca's area and 'b' Wernicke's area.

Where $t_{min}$ and $t_{max}$ are the minimum and maximum t-values, and N is the sum of left and right total number of voxels with positive t-values.

**Weighted histogram (histoLI).** Following the method reported by Branco, Suarez and colleagues [22,27], histograms of voxel counts versus t-value were determined for left and right ROIs. The t-values used for the histograms ranged from 0 to the maximal t-value within the ROI considered and were binned using an automatic binning algorithm, yielding bins with uniform width covering the whole range of t-values (MATLAB histcounts). A weighting function of squared t-values was then used to weight these voxel histograms [27]. The areas under the curves of left and right weighted histograms were calculated using a trapezoidal numerical integration, and compared as follows:

$$histoLI = (LA - RA)/(LA + RA) \qquad (5)$$

where LA and RA are the areas under the weighted histograms in left and right ROI, respectively.

**Area under the curve (AUCLI).** Another way of looking at the difference in activation in left and right ROI is to consider the number of voxels with values above the threshold for all possible thresholds [42]. This is achieved by setting every t-value present in the ROIs as a threshold and record the number of voxels with value above the threshold. The cumulative histograms of the obtained number of voxel vs threshold for left and right ROI (shown in Fig 2), can then be compared. The new method we propose in this work calculates the areas under these histograms and quantitatively compares them using the following formula:

$$AUCLI = (AUCL - AUCR)/(AUCL + AUCR) \qquad (6)$$

where AUCL and AUCR are the areas under the cumulative histogram calculated for the left

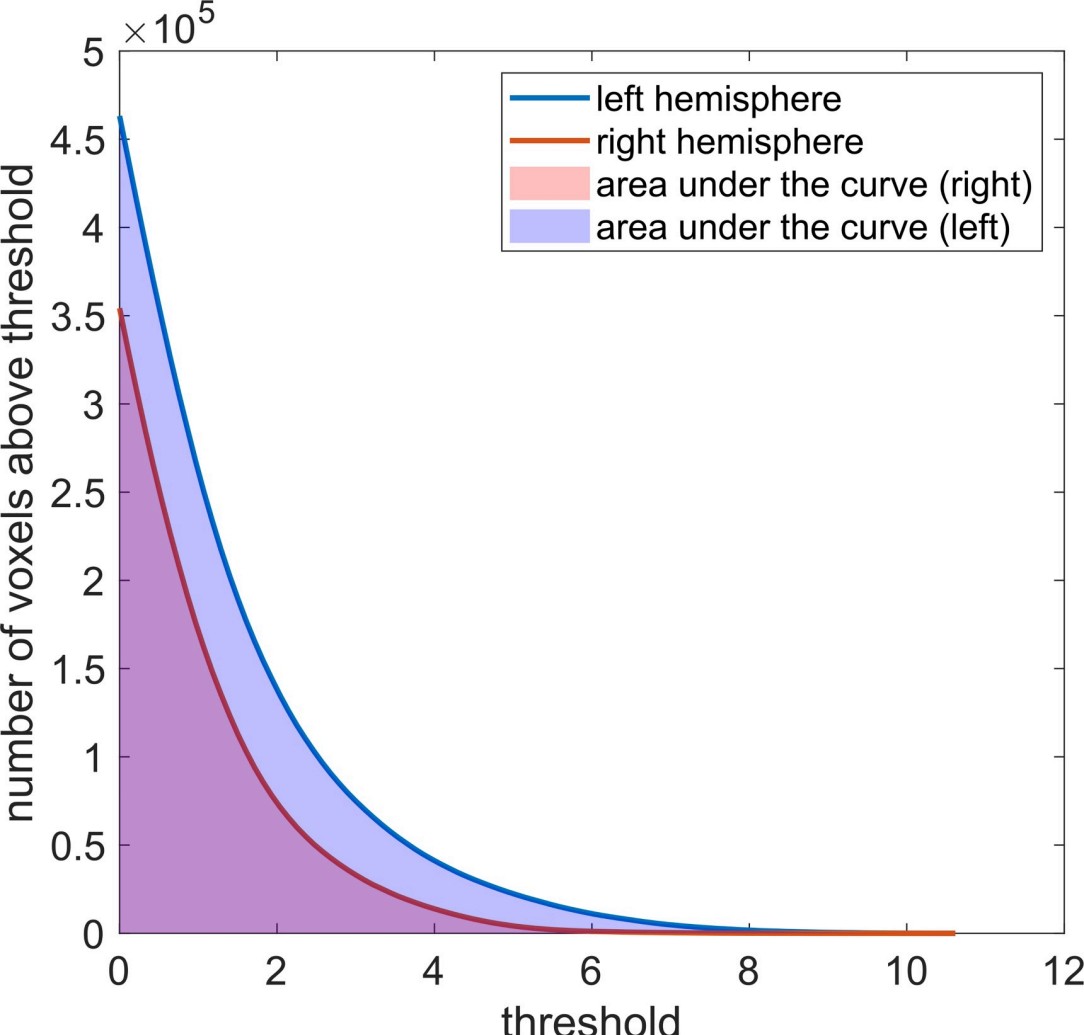

**Fig 2. Calculation of the laterality index with the AUCLI method.** The area under the cumulative histograms of number of voxels with value above threshold for left and right ROIs are used to compute a laterality index for the novel AUCLI method.

and right ROI, respectively. The areas under the cumulative histograms were calculated using the trapezoidal numerical integration available as a function in MATLAB.

## Method comparison

LI values were calculated for each fMRI dataset using the four different threshold-independent LI calculation methods. For each method, the subjects were ranked according to the calculated LI values, and pair-wise comparisons of rankings were performed for each task and ROI combination. To quantify the agreement between pairs of rankings, Spearman's correlation coefficients ρ were calculated using SPSS (Version 24, Statistics Software, IBM Corporation, USA).

## Results

All calculated LI values can be found as supplementary data in S1 Table.

LI versus threshold plots resulting from the conventional LI calculation are shown in Fig 3 for the three tasks and both ROIs. These plots demonstrate that the LI varies over the range of

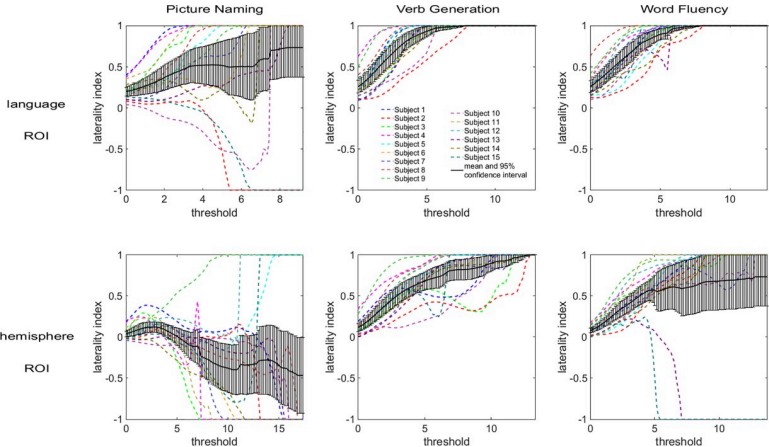

**Fig 3. Conventional laterality index plots.** The dependence of the laterality index on the threshold varies with the subject, the task performed, and the ROI chosen.

threshold, highlighting the existing issue of the conventional LI calculation, namely having to choose a single threshold value to evaluate lateralization. The LI versus threshold curves are not stable over the range of thresholds and can have sudden variations (drops), showing that the degree of lateralization can change considerably with the threshold. The 95% confidence interval of the mean LI is larger for the picture naming task than for the two other tasks, indicating that the differences between subjects are greater for this task. Furthermore, as both task and ROI have an impact on the LI distributions, the same subject can have different degrees of lateralization in different tasks or ROIs.

The dependence of the LI on the total number of activated voxels (curveLI method) also varies with subject, task, and ROI (see Fig 4). Nevertheless, the LI has smoother variations over the range of total number of activated voxels than over the range of thresholds, as expected [24]. Fig 4 shows variability between tasks similar to Fig 3: the spread in individual subject curves is greater for the picture naming task than for the two other tasks when considering the language ROI, and the lateralization of some subjects differs between tasks. Overall, the curves

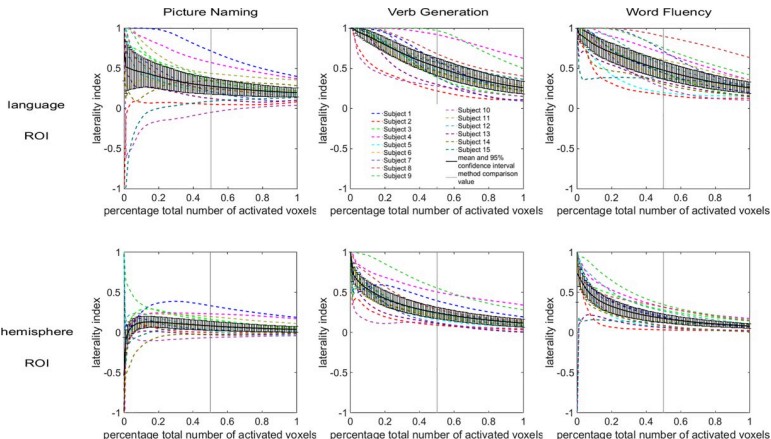

**Fig 4. Laterality index versus total number of activated voxels (curveLI) plots.** The dependence of the laterality index on the total number of activated voxels varies with the subject, the task performed, and the ROI. The data points for the mean LI and 95% confidence interval can be found as supplementary data in S2 Table.

obtained with the curveLI method show higher LI values for the language ROI than for the hemisphere ROI (see Fig 4). This indicates a more lateralized activation on a regional scale (language ROI) than on a larger scale (hemisphere ROI). However, within each task, the highest and lowest curves belong to the same subjects for both ROIs.

Fig 5 compares the inter-subject distributions of LI values in each method, for each task and ROI. The boxplots show higher LI values for the language ROI than for the hemisphere ROI in all methods. The median LI values obtained for the histoLI method are visibly discrepant from the other three methods. Fig 6 shows a different aspect of the same data, visually comparing the subject rankings obtained with each method. Within each graph, a color gradient (from blue for low LI values to red for high LI values) was applied, based on the subject ranking from the curveLI method (left side of the plot). In this visualization, crossing lines indicate differences in ranking between the methods. Fig 6 shows that, overall, similar subject rankings were obtained across the four methods, with Spearman's correlation coefficients ranging from 0.59 to 1.00 (Table 1). The pairwise comparisons involving the histoLI method resulted in the lowest Spearman's correlation coefficient. For the hemisphere ROI, the agreement between the AveLI and AUCLI methods is optimal in all tasks, while for the language ROI, the curveLI and AUCLI methods showed the best agreement in all tasks. Different tasks led to different subject rankings, but the level of agreement (i.e. range of Spearman's correlation coefficients) between methods in terms of subject ranking is similar between tasks. Within each task, there are some variations in the ranking between hemispheric and language ROIs, but weak or strong laterality is preserved for most subjects. The agreement between subject rankings obtained with the same method for different ROIs is good, with Spearman's correlation coefficients ranging from 0.75 to 0.96 (Table 2).

## Discussion

In this work, we compare four different threshold independent methods for assessment of language lateralization with fMRI. To eliminate the problem of the dependence of the LI on the statistical threshold, the different approaches adopt different metrics and it is therefore important to establish if, in practice, the choice of method has an impact on the assessment of language lateralization. A direct comparison by applying the methods to the same subject cohort represents a useful evidence base for the implementation of robust fMRI-based language lateralization in the clinical routine. This work has found that overall there is a good correlation between different methods in terms of subject rankings.

### curveLI method

The method based on choosing a fixed total number of activated voxels (curveLI) attempts to give an objective assessment of a patient's language lateralization in relation to a healthy control group [24], and is useful for looking at variations of the LI both within a subject and between subjects. Our results show that the curveLI method, compared to conventional LI assessment, reduced variability within subject. This matches the results from Abbott and colleagues, despite the differences in fMRI acquisition (MRI scanner, sequence parameters, tasks) and ROI definition between their study and ours, as well as our choice of normalization for the total number of activated voxels, highlighting the robustness of this method against the above-mentioned factors. However, there are visible differences between the mean LI value and 95% confidence interval obtained in our study for the word fluency task and those reported by Abbott and colleagues [24]. One limitation of this method is therefore that the reference set of curves might be specific to the local implementation and thus not transferrable to other datasets.

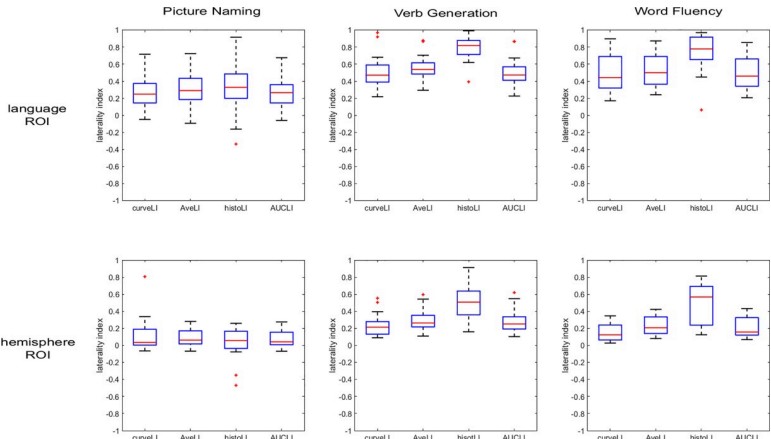

**Fig 5. Boxplots showing the variation between subjects within each method.** For each box, the central line indicates the median. The bottom and top edges indicate the first quartile $q_1$ and third quartile $q_3$, respectively. The whiskers extend to the most extreme data points not considered as outliers. Outliers are shown individually as '+' on the plots. Outliers are hereby defined as values larger than $q_3+w(q_3-q_1)$ or smaller than $q_1-w(q_3-q_1)$, where w is the maximum whisker length.

## AveLI method

The AveLI method produces a global laterality index value taking into account the lateralization at each activation value in the data series, giving more weight to voxels with high activation value compared to those with low activation [27]. The AveLI method has been shown to be resistant to outliers and stable against noise, to yield highly reproducible LI values between tasks, and to allow a good separation of subjects into left, right or bilateral language lateralization categories [27]. However, our results show that the LI values obtained with the AveLI method yield different subject rankings in different tasks. This might be due to differences in the type of tasks used. While we used picture naming, verb generation and word fluency tasks in English, Matsuo and colleagues used word generation and homophone judgement tasks in Chinese language.

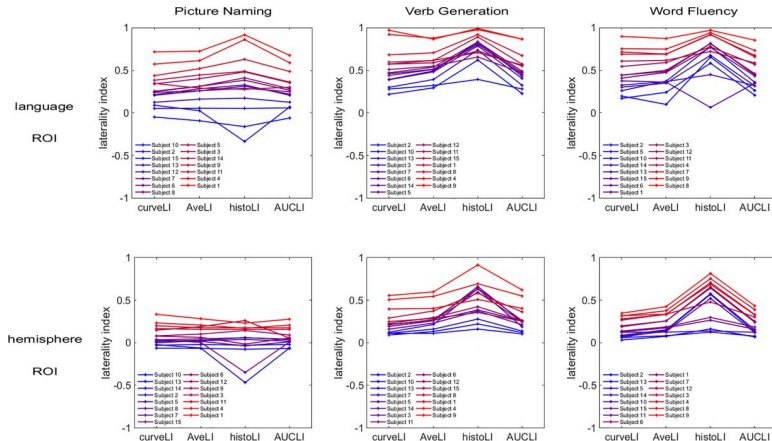

**Fig 6. Visual comparison of the subject rankings obtained with the different methods.** Within each graph, a gradual color change from blue for low LI values to red for high LI values is attributed to each subject, reflecting the subject ranking obtained with the curveLI method (left side of the plot). Crossing lines indicate differences in subject rankings between the methods.

**Table 1. Spearman's correlation coefficients for comparisons of subject rankings between the four different LI calculation methods for both ROIs.**

| | | picture naming task | | | | verb generation task | | | | word fluency task | | | |
|---|---|---|---|---|---|---|---|---|---|---|---|---|---|
| **language ROI** | | *curveLI* | *AveLI* | *histoLI* | *AUCLI* | | *curveLI* | *AveLI* | *histoLI* | *AUCLI* | | *curveLI* | *AveLI* | *histoLI* | *AUCLI* |
| | *curveLI* | 1.00 | 0.96 | 0.9 | 0.99 | *curveLI* | 1.00 | 0.99 | 0.68 | 1.00 | *curveLI* | 1.00 | 0.96 | 0.79 | 0.97 |
| | *AveLI* | 0.96 | 1.00 | 0.98 | 0.97 | *AveLI* | 0.99 | 1.00 | 0.65 | 0.99 | *AveLI* | 0.96 | 1.00 | 0.85 | 0.99 |
| | *histoLI* | 0.90 | 0.98 | 1.00 | 0.92 | *histoLI* | 0.68 | 0.65 | 1.00 | 0.71 | *histoLI* | 0.79 | 0.85 | 1.00 | 0.85 |
| | *AUCLI* | 0.96 | 0.97 | 0.92 | 1.00 | *AUCLI* | 1.00 | 0.99 | 0.71 | 1.00 | *AUCLI* | 0.97 | 0.99 | 0.85 | 1.00 |
| **hemisphere ROI** | | *curveLI* | *AveLI* | *histoLI* | *AUCLI* | | *curveLI* | *AveLI* | *histoLI* | *AUCLI* | | *curveLI* | *AveLI* | *histoLI* | *AUCLI* |
| | *curveLI* | 1.00 | 0.94 | 0.80 | 0.94 | *curveLI* | 1.00 | 0 98 | 0.59 | 0.96 | *curveLI* | 1.00 | 0.96 | 0.78 | 0.97 |
| | *AveLI* | 0.94 | 1.00 | 0.93 | 0.96 | *AveLI* | 0.98 | 1.00 | 0.63 | 0.99 | *AveLI* | 0.96 | 1.00 | 0.83 | 0. 99 |
| | *histoLI* | 0.80 | 0.93 | 1.00 | 0.86 | *histoLI* | 0.59 | 0.63 | 1.00 | 0.67 | *histoLI* | 0.78 | 0.83 | 1.00 | 0.84 |
| | *AUCLI* | 0.94 | 0.96 | 0.86 | 1.00 | *AUCLI* | 0.96 | 0.99 | 0.67 | 1.00 | *AUCLI* | 0.97 | 0.99 | 0.84 | 1.00 |

Similar subject rankings were obtained across the four methods for both ROIs, with Spearman's correlation coefficients ranging from 0.59 to 1.00.

## histoLI method

The reduced agreement between the histoLI method and the other three is likely due to the choice of the weighting function. We adopted a weighting function of squared t-values as recommended by Suarez and colleagues [28] for assessing language lateralization, instead of a linear weighting function as reported in their previous work looking at presurgical assessment of memory lateralization in the hippocampus [23]. This choice was motivated by the fact that language areas encompass a larger volume than the hippocampus [28]. In larger volumes, the probability of including voxels presenting low activation is higher so that a weighting function, which further increases the impact of voxels with high activation, is more appropriate. The weighting function of squared t-values reduces the influence of low t-value voxels (noise and false positives), which should improve the accuracy of the results, but also increases the influence of high t-value voxels. This overall results in a widened range of LIs, both towards the highly positive and the negative values (see Fig 5).

## AUCLI method

The novel method we propose in this work produces a single LI value evaluated over the entire range of activations present in the data series. The LI values calculated with the ACULI method are in good agreement with the other methods, especially with curveLI and AveLI. Nagata and colleagues [42] proposed another interesting LI calculation method. Their idea was to fit the same monomial equation to the number of activated voxels versus threshold curves for left and right ROI. The LI can then be calculated by comparing the fit parameters obtained for left and right ROI. When this method was applied to our data, it was not possible to find a common function that would satisfactorily describe both the left and right voxel histogram curves.

**Table 2. Spearman's correlation coefficients for comparisons of subject rankings between the language ROI and the hemisphere ROI for the four different LI calculation methods.**

| picture naming task | | | | verb generation task | | | | word fluency task | | | |
|---|---|---|---|---|---|---|---|---|---|---|---|
| *curveLI* | *AveLI* | *histoLI* | *AUCLI* | *curveLI* | *AveLI* | *histoLI* | *AUCLI* | *curveLI* | *AveLI* | *histoLI* | *AUCLI* |
| 0.75 | 0.84 | 0.85 | 0.78 | 0.96 | 0.91 | 0.75 | 0.88 | 0.90 | 0.86 | 0.90 | 0.85 |

The agreement between subject rankings obtained with the same method for the language ROI and the hemisphere ROI is good, with Spearman's correlation coefficients ranging from 0.75 to 0.96.

Both voxel histogram curves shown in Fig 2, could be fitted with an exponential or bi-exponential function but not with the same monomial function. A comparison of fit parameters to remove the direct dependence of the LI on the activation threshold as Nagata and colleagues [42] suggested, was thus not possible with such curves. The method we propose here is a simplification of Nagata's method: instead of comparing the two distributions using a fit function, we measure the area under the curves, which is independent of the shape of the curve. This approach also overcomes the drawback of having to find the best fit function, which may be specific to the task or ROI used.

## Suitability for the clinical routine

To be suitable for the clinical routine, a LI calculation should be 1) robust (i.e. independent of any parameter), 2) reproducible (i.e. stable over multiple calculations) and 3) allow an easy subject comparison [42]. The curveLI method and the AveLI method fulfil these three criteria. The histoLI method satisfies criteria 2) only for a determined weighting function, as acknowledged by Branco and colleagues [23] who first introduced this method. The AUCLI method evaluates the LI over the entire range of activation thresholds (t-values) present in the activation map, making it independent of the activation threshold and therefore robust. The AUCLI method also satisfies criteria 2) as the calculations of the cumulative histograms and areas under these will yield identical results each time performed, and therefore the final LI calculation will be stable over repeated calculations. All presented threshold-independent methods can provide a single summarizing LI value, which makes comparison between subjects easy, therefore satisfying criterion 3). However, the curveLI method also offers visual comparison of curves plotted in a single graph (Fig 4).Such a comparison is valid only if the number of activated voxels is appropriately normalized (for example to the total number of voxels with positive values within the SPM for each subject, as done in this work), or if the total number of positive voxels within the ROI considered is the same for all subjects, as done by Abbott and colleagues [24]. When a reference cohort is available, the curveLI method offers an easy visual comparison (direct assessment of where the patient curve lies in respect to the mean LI curve and 95% confidence interval), while retaining the more complex information of the smooth dependence of the the LI on the total number of activated voxels. In addition to the three criteria listed by Nagata and colleagues [42], 4) ease of implementation and 5) speed of data analysis are important for use in the clinical routine. All four methods investigated in this study require custom scripts, but the implementation of the AUCLI method is more straightforward as the metric used is simpler than in the other methods. For a single subject, the LI calculation required approximatively 10 seconds with the curveLI method, 60 seconds with the AveLI method, 5 seconds with the histoLI method, and 1 second with the AUCLI method using our computer environment (Mac OS X El Capitan Version 10.11.6, Processor 3.3 GHz Intel Core i5, RAM 32 GB 1867 MHz DDR3). Taking these five criteria into account, the novel method proposed in this work can be considered suitable for the clinical routine and presents advantages compared to the previously reported methods.

## Choice of ROI

In this work, two different ROIs have been considered, giving the possibility to assess language lateralization on both a hemispheric scale and a more regional scale. Even though the agreement between methods is slightly higher when the language ROI is considered rather than the hemisphere ROI, choosing the language ROI defined using standard brain atlases might not be appropriate in clinical subjects. Problems with the registration of the MNI standard brain to the acquired brain might arise due to abnormal brain anatomy. Furthermore, the actual

functional area of the patient might not correspond to the atlas-based language ROI even if the registration yields satisfactory results. In patients with abnormal anatomy and potentially displaced functional centers, it might thus not be possible to accurately define functional ROIs based on anatomical or functional a-priori knowledge. In such cases, which are common in presurgical assessment of language lateralization, larger ROIs encompassing the whole hemisphere may be preferable at the sacrifice of including areas that are not associated with language. For instance, strong activation was visible in the visual cortex for all subjects as a result of the visual stimuli. This activation is not perfectly symmetric and can thus influence the calculated LI values. This problem could be overcome by masking out the visual cortex to exclude this area of the brain from the calculation and remove its influence on the LI values or by modifying the paradigm design to obtain similar visual activation during resting and active part of the task. Another drawback of using a hemispheric ROI to assess language lateralization is the impossibility to describe the differences in lateralization between different regions within the brain hemispheres. Regional language lateralization has been shown to differ from hemispheric language lateralization in both healthy subjects and patients [6]. The information about a regional lateralization might be of greater interest than a hemispheric lateralization in case of a very localized lesion. The choice of hemispheric or regional ROI should thus be made according to the patient's condition.

## Choice of language tasks

In this work, we have considered three language tasks. The results show that the assessment of language lateralization is task dependent, confirming previous conclusions [43–45]. This is a result of the complexity of human language, which involves numerous different functions [1,2,45]. Not all language functions can be assessed by a single fMRI task and it is therefore recommended to use an assembly of tasks to assess language lateralization more accurately [2,5,45]. The LI values calculated for the picture naming task are noticeably lower than those obtained for the verb generation and word fluency tasks (see Fig 6). Our results thus confirm that the picture naming task is less reliable than the two other tasks considered in this work. However, the picture naming task is simpler to perform and may therefore be more appropriate for patients with cognitive deficits [45].

## Limitations

The multilingual character of the cohort of healthy subjects used in this project influences the range of LI values obtained since activation patterns have been shown to differ in native speakers and non-native speakers [46], potentially leading to a more spread out confidence interval of the mean LI values. However, a multilingual control group might be more representative of the expected patient population in certain hospitals.

The inherent limitation of fMRI resulting from poor patient cooperation and task performance should always be kept in mind. Especially when using silent language tasks, it is very difficult to ensure the patient is performing the task properly during the scan. Therefore, it is important to give clear instructions before the scan and some practice outside the scanner might even be advisable. For patients with cognitive deficits it may be necessary to verify whether the tasks can be performed before the scanning session. Some adaption of the task design (e.g. color of writing and background, display during rest period, frequency of image/word/letter) might also be useful for clinical cases to improve the patient's ability to perform the task.

This work based on volunteer data, was part of the process to establish fMRI-based presurgical assessment of language lateralization in the clinical routine at our hospital, and is focused

on the comparison of different LI calculation approaches. Healthy volunteer data, in addition to provide a reference cohort, is ideally suited for direct method comparison as they are independent of differences in patient pathologies. As patients are now being considered for presurgical assessment of language lateralization, future developments of this work will be to compare the methods reported here in clinical subjects, and further investigate the choice of hemispheric vs language ROI. In these subjects, comparison of the fMRI results to direct electrical stimulation during surgery–the gold standard for determining lateralization–will also be possible.

## Conclusion

For robust assessment of language lateralization in the clinical routine, it is advisable to use a threshold-independent laterality index calculation. In this work, we have tested four different methods on the same subject cohort. Our results show that the choice of method itself is not key, as all methods agree in differentiating strong from weak lateralization on both hemispheric and regional scales. This choice should nevertheless be consistent to allow a relative comparison of language lateralization between subjects. Our results highlight that the laterality index is not an absolute measure, as numerous factors—some purely related to the LI calculation method—can influence its value. In this work, we have introduced a new threshold-independent laterality index calculation method and validated it against three previously reported methods. Our evaluation suggests that the new method is well suited for application to clinical practice as it is simple to implement, fast, robust, reproducible, and allows direct subject comparison.

## Supporting information

**S1 Table. LI values calculated for all subjects with the four threshold-independent LI calculation methods for all tasks and both ROIs.**
(XLSX)

**S2 Table. Mean LI value and 95% confidence interval data points obtained with the curveLI method.**
(XLSX)

## Author Contributions

**Conceptualization:** Irène Brumer, Enrico De Vita, Jonathan Ashmore, Jozef Jarosz, Marco Borri.

**Data curation:** Irène Brumer, Marco Borri.

**Formal analysis:** Irène Brumer.

**Funding acquisition:** Enrico De Vita, Jozef Jarosz.

**Investigation:** Irène Brumer, Marco Borri.

**Methodology:** Irène Brumer, Marco Borri.

**Project administration:** Marco Borri.

**Resources:** Jozef Jarosz, Marco Borri.

**Software:** Irène Brumer, Jonathan Ashmore.

**Supervision:** Enrico De Vita, Marco Borri.

**Validation:** Irène Brumer, Marco Borri.

**Visualization:** Irène Brumer.

**Writing – original draft:** Irène Brumer.

**Writing – review & editing:** Enrico De Vita, Jonathan Ashmore, Jozef Jarosz, Marco Borri.

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
