## [Decision Letter · Decision Letter 0]

12 Nov 2019

PONE-D-19-25144

Implementation of clinically relevant and robust fMRI-based language lateralization: choosing the laterality index calculation method

PLOS ONE

Dear Dr De Vita,

Thank you for submitting your manuscript to PLOS ONE. After careful consideration, we feel that it has merit but does not fully meet PLOS ONE’s publication criteria as it currently stands. Therefore, we invite you to submit a revised version of the manuscript that addresses the points raised during the review process.

The reviewers suggested a number of clarifications and extensions of the manuscript, please respond as required.

We would appreciate receiving your revised manuscript by Dec 27 2019 11:59PM. To enhance the reproducibility of your results, we recommend that if applicable you deposit your laboratory protocols in protocols.io, where a protocol can be assigned its own identifier (DOI) such that it can be cited independently in the future. For instructions see: http://journals.plos.org/plosone/s/submission-guidelines#loc-laboratory-protocols

We look forward to receiving your revised manuscript.

Kind regards,

Peter Lundberg

Academic Editor

PLOS ONE

Journal Requirements:

2. In your data availability statement you write all the relevant data are within the paper and/or its Supporting Information files. Please ensure you have provided the individual data points used to create the figures and determine means, medians and variance measures presented in the results, tables and figures (http://journals.plos.org/plosone/s/data-availability#loc-faqs-for-data-policy). If these data cannot be publicly deposited or included in the supporting information, e.g. due to patient privacy or ownership by a third party, explain why and explain how researchers may access them.

Reviewers' comments:

Reviewer's Responses to Questions

**Comments to the Author**

1. Is the manuscript technically sound, and do the data support the conclusions?

Reviewer #1: Yes

Reviewer #2: Yes

2. Has the statistical analysis been performed appropriately and rigorously? 

Reviewer #1: Yes

Reviewer #2: Yes

3. Have the authors made all data underlying the findings in their manuscript fully available?

Reviewer #1: No

Reviewer #2: No

4. Is the manuscript presented in an intelligible fashion and written in standard English?

Reviewer #1: Yes

Reviewer #2: Yes

5. Review Comments to the Author

Reviewer #1: Thank you for a well-written article on an interesting topic, keeping the useability of the results for clinicians and researchers in mind throughout the paper. I have some minor comments.

I cannot find the full dataset (fMRI raw data or statistical parameter maps) provided anywhere as attachment, supplementary data or link to data repository.

Line 168/169 on page 8 would read better if the numbers and letters referred to were put in quotation marks, e.g. "... with 'a' designating Broca's area"

The description of the novel method, AUCLI, was lacking some details that would aid reproducibility of the method. First, line 213-215 reads "... count the number of voxels with values above the threshold for all t-values present in the ROIs". Is this meant to apply to each threshold present in the ROIs? The sentence reads as if one threshold is set, but this cannot be the case as the manuscript is advocating the use of threshold-independent LIs.

The first sentence of the Results, at page 11 line 239 refers to LI versus threshold plots that are likely referring to the conventional LI. Please do state this more clearly in this sentence. As LI is not consistently used only to refer to the conventional LI calculation, it is helpful to be very consistent in naming. Likewise, in the description of Fig 4 it is helpful to refer to the in-manuscript name of the method; curveLI.

In the figure text of Fig 5 (page 13 lines 295-296) there's a reference to outliers. The manuscript is lacking a clear description of outlier detection chosen, and if there is need for outlier detection (rationale, implications).

At page 17/18 there is not a clear positive answer to whether other methods than CurveLI adhere to criterion 3 for suitability for clinical implementation: "allow an easy subject comparison". Therefore, the conclusion drawn in line 412/413 "Taking these five criteria into account, the novel method proposed in this work can be considered suitable for the clinical routine" cannot be drawn which undermines one of the main findings of the manuscript. Of course, the authors do show means of comparing between subjects throughout the manuscript, so this should be described in this section together with a judgment of whether this is adequate enough to be deemed as "easy", or else rephrase their conclusions regarding clinical implementability.

It should be noted that the colors and gradient used in Fig 6 are virtually indiscriminable when viewed as a black and white printout, it would be advisable to switch to colors with a clear difference in b/w.

Reviewer #2: The authors present a study comparing methods for calculating laterality index. While the findings may not have broad appeal, they will certainly be of use for groups performing clinical fMRI for pre-surgical mapping.

I have a number of comments, though.

1. Why was such a slow TR of 3 seconds chosen? It has been apparent for a while that the sensitivity of fMRI increases with decreasing TR.

2. When using SPM, it is generally recommended to choose a smoothing kernel that is approximately 2x the size of the acquired voxel (see Friston et al., 1996 NeuroImage; Ball et al., 2012 Human Brain Mapping; Pajula and Tohka 2014, MRI; Liu et al., 2017, J Neuroscience Methods). Given that the acquired voxel is 2.5x2.5x3mm, can the authors justify using an 8mm smoothing kernel, which is almost 3x the size of the acquired voxel?

3. What did the authors include in their GLM beyond the task regressors? Were the realignment parameters estimated during the realignment step included as covariates of no interest? What steps were taken to mitigate the effect of noise in the data? What quality checks were performed on the MRI data? Individual examination for excessive head motion?

4. I do not understand the purpose of Figure 6. What is it showing that could not be shown in Figure 5, if the authors included the individual data points as part of the box and whisker plots in Figure 5? The authors need to better explain what is being shown in Figure 6, especially why it is important to color code the subjects based on curveLI method?

5. Did the authors consider acquiring multiple runs of each task in a single session or performing multiple scanning sessions to enable any sort of test-retest or cross-validation of the results? This would be an interesting result to see in terms of how robust each LI method is.

6. In the conclusions, the authors mention that laterality index measures can be influenced by post-processing method. Given the results presented in the manuscript, how did the authors arrive at this conclusion? I did not see any indication that the authors varied anything but whether LI was calculated at the hemispheric level or using language-related ROIs. To justify this conclusion, I would have expected to see the authors comparing smoothing kernels, standard software packages/pipelines, the inclusion/exclusion of noise confounds in the GLM, etc.

7. The authors have omitted many of the required references and acknowledgments for using the Harvard Oxford and Julich Histological atlases (https://fsl.fmrib.ox.ac.uk/fsl/fslwiki/Atlases).

6. PLOS authors have the option to publish the peer review history of their article (what does this mean?). If published, this will include your full peer review and any attached files.

Reviewer #1: No

Reviewer #2: No

---

## [Author Response · Author response to Decision Letter 0]

17 Dec 2019

A full response to reviewers comments is attached as a separate file

---

## [Decision Letter · Decision Letter 1]

24 Feb 2020

Implementation of clinically relevant and robust fMRI-based language lateralization: choosing the laterality index calculation method

PONE-D-19-25144R1

Dear Dr. De Vita,

We are pleased to inform you that your manuscript has been judged scientifically suitable for publication and will be formally accepted for publication once it complies with all outstanding technical requirements.

With kind regards,

Peter Lundberg

Academic Editor

PLOS ONE

Additional Editor Comments (optional):

Reviewers' comments:

Reviewer's Responses to Questions

**Comments to the Author**

1. If the authors have adequately addressed your comments raised in a previous round of review and you feel that this manuscript is now acceptable for publication, you may indicate that here to bypass the “Comments to the Author” section, enter your conflict of interest statement in the “Confidential to Editor” section, and submit your "Accept" recommendation.

Reviewer #1: All comments have been addressed

Reviewer #2: All comments have been addressed

2. Is the manuscript technically sound, and do the data support the conclusions?

Reviewer #1: Yes

Reviewer #2: Yes

3. Has the statistical analysis been performed appropriately and rigorously? 

Reviewer #1: Yes

Reviewer #2: Yes

4. Have the authors made all data underlying the findings in their manuscript fully available?

Reviewer #1: Yes

Reviewer #2: Yes

5. Is the manuscript presented in an intelligible fashion and written in standard English?

Reviewer #1: Yes

Reviewer #2: Yes

6. Review Comments to the Author

Reviewer #1: Thank you for your clarifications, all my concerns have been adequately addressed.

I would advice to include a statement about the described restrictions in data sharing in the article.

Also, note that the word 'histograms' is accidentally written twice in line 217

Reviewer #2: (No Response)

7. PLOS authors have the option to publish the peer review history of their article (what does this mean?). If published, this will include your full peer review and any attached files.

Reviewer #1: No

Reviewer #2: No

---

## [Editor Report · Acceptance letter]

2 Mar 2020

PONE-D-19-25144R1 

Implementation of clinically relevant and robust fMRI-based language lateralization: choosing the laterality index calculation method 

Dear Dr. De Vita:

I am pleased to inform you that your manuscript has been deemed suitable for publication in PLOS ONE. Congratulations! Your manuscript is now with our production department. 

With kind regards,

on behalf of

Professor Peter Lundberg 

Academic Editor

PLOS ONE